# Simulation of RSO Images for Space Situation Awareness (SSA) Using Parallel Processing

**DOI:** 10.3390/s21237868

**Published:** 2021-11-26

**Authors:** Ryan Clark, Yanchun Fu, Siddharth Dave, Regina Lee

**Affiliations:** Department of Earth and Space Science, York University, 4700 Keele Street, Toronto, ON M3J 1P3, Canada; ycf@my.yorku.ca (Y.F.); sgdave@yorku.ca (S.D.); reginal@yorku.ca (R.L.)

**Keywords:** space situational awareness (SSA), resident space objects (RSOs), artificial intelligence (AI), parallel processing

## Abstract

With the rapid increase in resident space objects (RSO), there is a growing demand for their identification and characterization to advance space simulation awareness (SSA) programs. Various AI-based technologies are proposed and demonstrated around the world to effectively and efficiently identify RSOs from ground and space-based observations; however, there remains a challenge in AI training due to the lack of labeled datasets for accurate RSO detection. In this paper, we present an overview of the starfield simulator to generate a realistic representation of images from space-borne imagers. In particular, we focus on low-resolution images such as those taken with a commercial-grade star tracker that contains various RSO in starfield images. The accuracy and computational efficiency of the simulator are compared to the commercial simulator, namely STK-EOIR to demonstrate the performance of the simulator. In comparing over 1000 images from the Fast Auroral Imager (FAI) onboard CASSIOPE satellite, the current simulator generates both stars and RSOs with approximately the same accuracy (compared to the real images) as STK-EOIR and, an order of magnitude faster in computational speed by leveraging parallel processing methodologies.

## 1. Introduction

Space Situational Awareness (SSA) is becoming an increasingly important issue around the world as the number of resident space objects (RSOs) is continually increasing. SSA requires innovative, robust, and reliable solutions to identify, track, and characterize RSOs. Previously, we demonstrated the feasibility of using low-resolution on-orbit images (such as star tracker images) for RSO detection. In [1], we examined the number and frequency of RSOs that are detectable (given the physical and optical parameters of the objects and imager such as distance, brightness, motion) when a commercial-grade star tracker is used instead of dedicated high-resolution imagers. Using the simulation study [1], we demonstrated that hundreds of objects in low earth orbit can be observed using a star tracker in a day. The starfield simulator—named space-based optical image simulator (SBOIS)—developed for this feasibility study was modeled after FAI imager onboard CASSIOPE spacecraft. Similar simulators; such as the ones outlined in [2,3] use a propagation and ray-tracing software where SBOIS performs all calculations in MATLAB with no additional software required. The simulation represents low-resolution star-tracker-like images invisible to IR range. More information on the preliminary starfield image simulator is also provided in [4]. SBOIS also serves as the training tool to enable artificial intelligence (AI) algorithm design to identify and characterize the RSOs from low-resolution images. AI algorithms have been shown in recent years to perform accurately and efficiently after training, such as [5] for RSO detection and [6] for RSO characterization. Research such as [7,8] highlights the need for high quality and quantity of training data about starfield images and corresponding labels. A labeled starfield database, as produced by SBOIS, enables novel model development for resolving imagery for classification purposes. As detailed in [9] machine learning algorithms adapted to classification and characterization can leverage large dataset variations which are only achievable via tools like SBOIS. Therefore from the perspective of image processing algorithms, the key-value metrics of an image simulator like SBOIS are labeled image generation accuracy, variability in simulation, and image generation time.

While SBOIS provides powerful means to generate and simulate a large number of images, the simulator’s performance was limited in its computation efficiency, as well as, the versatility in providing images from multiple sources. In this paper, we present the next-generation starfield simulator design that features object centroid estimation; implementation of parallel processing for optimal computation; and robustness of input simulation parameters. To validate the accuracy of the simulator, we compared the resulting simulation of the starfield images to the actual images obtained from FAI as well as the simulated images from the Systems Tool Kit (formerly Satellite Tool Kit, STK) a commercially available application.

The accuracy and efficiency of the simulation in starfield images is a critical step in developing SSA algorithms. In [10], the importance of ‘high-fidelity and innovative simulation architecture’ is described to engage SSA mission design. PROXORTM/RT- PROXORTM by Bell Aerospace is one of the few commercially available starfield simulators using multi-thread architecture. STK offers an alternative tool to visualize and analyze stars and RSOs in orbit. For the purpose of this study, we compared the proposed simulator design to STK with electro-optical and infrared sensors (EOIR) toolkit. Originally developed by the Space Dynamics Laboratory for missile defense, STK-EOIR supports radiometric sensor modeling of optical sensors. Combined with STK’s various mission design capabilities, STK-EOIR is a unique platform where optical RSO tracking missions can be planned, simulated, and studied with a graphical user interface for convenience. More details of STK-EOIR features are described in [11]. Ref. [12] presents a study on space object identification using deep neural networks using STK-EOIR where the photometric observations are generated from STK-EOIR with the GEODSS sensor. While STK-EOIR proves to be a versatile platform to simulate starfield images for mission planning and proof-of-concept demonstration, it still lacks the flexibility and efficiency we seek for the current studies applications. We recognize that a commercial product like STK offers a unique capability to perform complex analyses as well as training and research opportunities. AGI reports that STK has more than 50,000 installations at more than 800 global organizations. While limited STK functionalities are available for free, advanced functions still require expensive licensing. Furthermore, for the purpose of the current study, flexibility in low-level implementation is required to accommodate the large set of data and multiple sensor platforms. As such, a custom starfield simulator specifically designed for the RSO identification study was designed and based on the simulator we had developed for the feasibility study. In developing SBOIS, three key parameters were considered to enable RSO tracking algorithms: (1) RSO centroid estimation; (2) implementation of parallel processing; and (3) versatility in simulation parameters.

## 2. Simulator Architecture

SBOIS was originally designed to examine the feasibility of RSO detection in space. To understand the characteristics of on-orbit images with RSOs, we simulated various scenarios based on commercial grade star trackers currently available in the market. We considered several star trackers, including AD-1 Star Tracker by Mars Bureau [13], and the BOKZ-MF Star tracker by IKI RAN [14]. All models are great candidates for star imaging on a small satellite platform for attitude determination. However, for the purpose of RSO detection, we focused on Fast Auroral Imager (FAI) parameters as the baseline imager, due to a large number of images being publicly available with well-known host satellite positions. The FAI sensor is onboard the CASSIOPE satellite as part of the ePOP (Enhanced Polar Outflow Probe) science mission. The purpose of FAI is to measure large-scale auroral emissions. It measures auroral emissions with near-infrared wavelengths (650–1100 nm) and a monochromatic wavelength of 630 nm. The near-infrared range was chosen instead of the chromatic wavelength as it is a more realistic representation of low-resolution images like what would be found from star tracker images. The parameters for FAI are shown in Table 1. Images from FAI are publicly available on the ePOP website [15]. FAI sensor has an effective f-number of f/0.8, an effective focal length of 13.8 mm. SBOIS simulates the optical environment from various sources of direct and reflected flux from sources that include; the starfield, Earth, Earth Limb, Moon, Sun, and Zodiac. To combine these different flux sources, as well as, the RSO/s into one image the architecture in Figure 1 is used. SBOIS uses the common assumption that the RSOs are not active sources of flux but rather, reflectors of solar radiation, which is common in recently published RSO models [16,17,18]. To account for the active sources of flux, they are considered either as point sources or area sources with different calculations performed for each. Point sources include the Sun, and stars that are calculated from their known magnitude. The equations to calculate the given magnitude to the Digital Number (DN) values are shown in Equations (1) and (2) [18].
*E_Obj_* = *E_zero_* × 10^−0.04*m_obj_*^(1)
*DN_x,y_* = *E_obj_* × *eDN* × *QE* × *τ* × *A_app_* × *t_int_* × *PSF*(*x*, *y*, *x**_obj_*, *y**_obj_*)(2)

In Equations (1) and (2): *E_Obj_* represents the photon irradiance from the objects in photons per second per meter squared ph/s/m^2^. *E_zero_* represents the zero-point photo irradiance found from integrating the solar spectrum over the effective band, in ph/s/m^2^. *m_obj_* represents the absolute magnitude of the object. *DN_x_*_,*y*_ represents the Direct Number (DN) value at pixel *x* and *y* on the sensor. *eDN* represents the ratio of photons received to the electrons given by the sensor properties, the units are electrons per DN, *e*/*DN*. *QE* represents the quantum efficiency of the sensor system in the band used. *τ* represents the optical feed loss of the sensor. *A_app_* represents the area of the aperture given in *m*^2^. *t_int_* represents the integration time of the sensor in seconds. *PSF*(*x*, *y*, *x_obj_*, *y_obj_*) represents the point spread function on pixel *x*, *y* for a source that is located at *x_obj_*, *y_obj_*. For more information on point spread functions and the function implemented please see [1]. Note that the Sun is never included in the image due to the Sun exclusion angle.

For area sources such as Earth, Earth’s limb, the Moon, Moonglow, and zodiac glow are calculated from their surface brightness. The equations to transform the given magnitude to the DN values are shown in Equations (3)–(5).
(3)$$L_{obj}=E_{zero}\times 10^{-0.04m_{asc,\;obj}}{\left(\frac{180}{\pi }\right)}^{2}3600^{2}$$
(4)$$E_{obj}=L_{obj}\times IFOV^{2}$$
(5)$$DN_{pp}=E_{obj}\times eDN\times QE\times \tau \times A_{app}\times t_{int}$$

In Equations (3)–(5): *L_obj_* represents the photon radiance given in units of photons per second per meter squared per steradian ph/s/m^2^/sr. *E_zero_* represents the zero-point photo irradiance found from integrating the solar spectrum over the effective band, in units of ph/s/m^2^. *m_asc_*_,_ *_obj_* represents the magnitude per square arc-second of the surface object given in magnitude per arc-second squared. *E_Obj_* represents the photon irradiance from the objects in photons per second per meter squared, ph/s/m^2^. *IFOV* represents the instantaneous field of view which is found by dividing the pixel size by the focal length. *DN_pp_* represents the DN values per pixel across the sensor. *eDN* represents photoelectron the DN ratio given in electrons per DN, *e*/*DN*. *QE* represents the quantum efficiency of the sensor system in the band being used. *τ* represents the optical feed loss of the sensor. *A_app_* represents the area of the aperture given in units meters squared m^2^. *t_int_* represents the integration time of the sensor in seconds. Note that Equation (3) represents the conversion from arcseconds squared to degrees squared, with ${\left(\frac{180}{\pi }\right)}^{2}$ representing the conversion from degrees squared to steradians.

To calculate the brightness of the RSO’s in the image, a Bidirectional Reflectance Distribution Function (BRDF) is used with the incoming solar flux. To perform the BRDF modeling of RSO’s, we assume that satellites consist of several well-defined surfaces called facets. These facets are surfaces with well-known BRDF’s. Complex 3-dimensional shapes consist of these facets transforming the complex and unknown BRDF function of RSO’s into a sum of its known facets. In the current study, all RSO’s are assumed to be perfectly spherical objects with a 0.2 reflectance value, as this is the simplest case to determine detections. Equations (6)–(11) show the calculation of DN values from the spherical facet model of RSO’s. More details on the BRDF and facet modeling of SBOIS are described in [19].
(6)$$\rho AF=\rho A\left(B_{diff}F_{diff}+B_{spec}F_{spec}\right)$$
(7)$$F_{diff,Spherical}=\frac{2}{3\pi^{2}}\left[\left(\pi -\Phi \right)\cos\left(\Phi \right)+\sin\left(\Phi \right)\right]$$
(8)$$F_{spec,Spherical}=\frac{1}{4\pi^{2}}$$
(9)$$m_{obj}=m_{sun}-2.5\log_{10}(\frac{\rho AF}{a^{2}}\;)+5\log_{10}(\frac{d_{rso,sun}\times d_{rso,sensor}}{a^{2}}\;)$$
(10)$$E_{Obj}=E_{zero}\times 10^{-.04m_{obj}}$$
(11)$$Dn_{x,y}=E_{obj}\times eDN\times QE\times \tau \times A_{app}\times t_{int}\times PSF\left(x,y,x_{obj},y_{obj}\right)$$

In Equations (6)–(11), *ρAF* represents the reflective area phase function which is an intermittent step on calculating the magnitude of the RSO. *ρ* represents the reflectivity of the object. *A* represents the effective area of the RSO in meters squared. *B_di f f_* and *B_spec_* represent the diffuse and specular Hejduk mixing coefficients, respectively [17]. *F_di f f_* and *F_spec_* represent the diffuse and specular phase function, respectively. Φ represents the solar phase angle of the Sun, RSO, sensor system. *m_obj_* represents the magnitude of the RSO. *m_sun_* represents the magnitude of the sun, which is constant at −26.73. *a* represents one astronomical unit, the distance from the Sun to Earth. *d_x_*_,*y*_ represents the distance from object x to object y. The rest of the variables are the same as in Equations (1) and (2) which are described above.

After all the sources of flux are included in the image various noise are considered and added including, shot noise, dark current, read noise, multi-pixel radiation events, and the point spread function. For more information on the noise modeling of SBOIS_164_ see [4].

The final stage of the simulator combines all the flux sources and noise to create an image with the labeled information outputted as auxiliary data with the image. One powerful feature of SBOIS is the ability to simulate different types of RSO’s, such as active and inactive, that have different types of auxiliary data available. This allows for robustness in the inputs allowing accurate ephemeris data to be entered for satellites with well-known positions, or TLE propagation for RSO’s with less accurately known orbits. Robustness in algorithms generally leads to a decrease in either computation speed or accuracy. By leveraging parallel processing capabilities in the input, RSO propagation, and image generation, the overall computation speed can be reduced while keeping the robustness of the inputs. One of the challenges in developing a robust RSO image simulator is the different types of active and inactive RSO’s, which contain fidelity of auxiliary data. SBOIS is designed to generate a robust image simulator, accepting multiple types of data inputs, as well as, being more efficient than currently available image simulators by leveraging parallel processing capabilities.

The robustness of the inputs was designed with the assumption that different types of observations and RSO’s have different levels of information available. Active satellites would have the most information available, normally having ephemeris data, attitude, and general shape information available. The ephemeris data allows for better positioning of the target RSO compared with propagation methods. The attitude and general shape allow for a more accurate estimate of the facet model of the satellite allowing the light reflected off the RSO to be simulated with more accuracy. This information provides the most accurate simulation with SBOIS but is not always available for all RSO’s. For deactivated satellites and rocket bodies, ephemeris data and attitude are not always available. When the ephemeris data is not available a TLE and PSGP4, mentioned in more detail below, are used to determine the position of the RSO. With no attitude information, and in the case of debris where the shape is sometimes not known, the RSO’s are simulated as a perfectly reflective sphere. This is commonly done in different RSO simulators to estimate the effective light coming off the object which allows for the training and evaluation of detection algorithms [5], as well as detection prediction [1]. The robustness of the outputs was also designed to provide any information that the user requires as auxiliary data to allow for easy training of machine learning algorithms. These outputs include; coordinate rotation matrices, object brightness, labeled star location, and most importantly labeled RSO information. The labeled RSO information gives the RSO’s that are visible in the image with their position, relative motion, relative magnitude, and signal to noise ratio. This allows for the SBOIS simulated images to be used in the training of the detection algorithms without the need for any external labeling code/software.

## 3. Parallel Processing and PSGP4 Architecture

Image simulation is a computationally intensive task, this is only increased with the simulation of thousands of moving RSO’s possibly in the image, background zodiac light, moon glow, etc. The impact of this is requiring seconds to minutes to generate one simulated image using standard sequential computation methods. To allow for large data sets (thousands to millions of images) to be generated in a reasonable time, the computation time per image needs to be reduced when looking at commercially available simulators such as STK’s EOIR. To accomplish this parallel processing in the form of Single Instruction Multiple Data (SIMD) parallel processing is introduced. SIMD leverage the improvements in computer hardware allowing a single instruction to be performed on multiple sets of data at once, reducing the computation time required to perform the same calculation on multiple data points. For image simulation, this is commonly used with ray-tracing algorithms performing the same calculation on thousands to millions of rays. The SBOIS uses a forward ray tracking method with the different sources of light mentioned above. The implementation of SIMD allows for the light received from different sources, as well as, all the light reflected off of RSO’s to be performed at once greatly reducing the computation time required. SIMD cannot be implemented on every part of SBOIS, though it was able to be implemented on the most computationally intensive steps which include; coordinate transformations, received flux, propagation, and assigning values to pixels.

The simulation of RSO images requires the known position of the RSO’s at the given time an image is taken. As for most RSO’s ephemeris data is not available TLE’s are used to propagate an RSO to approximately its position in the image. This has many uses such as estimating the distance to the RSO, seeing if the RSO is in the Field of View (FOV), and determining the pixel location of the RSO if it is in the FOV. Propagation is one of the more computationally intensive portions of simulating RSO images, this scales with the amount of RSO’s propagated. For an image where the RSO is not known propagation of an entire catalog is required to determine what RSO is in the image. Currently, there are approximately 20,000 tracked objects in the NORAD Satellite Catalog, with that number expected to jump an order of magnitude over the next decade [19]. Due to the proliferation of tracked RSO’s more efficient methods for satellite propagation are required to generate large datasets efficiently. This need has led to the creation of the Parallel Specialized Perturbation Method 4 (PSGP4) Propagator.

PSGP4 is a version of the SGP4 propagator that is set up to propagate RSO’s in parallel. Due to the propagation of RSO’s following almost the same architecture and having no correlation between different RSO’s, this allows for the implementation of SIMD parallel processing. This is shown from the algorithm block diagram, Figure 2. While there are some sections where SIMD is not possible, such as the solving of Kepler’s equation which is performed iteratively, it is implemented in a significant portion to see improvements in computation time without a reduction in accuracy. This can be seen from the implementation of PSGP4 for maneuver detection in Parallel Propagation for Orbital Maneuver Detection [20]. Since the paper was published in 2020 the PSGP4 algorithm has been updated to include the Specialized Deep Space Perturbation Method 4 (SDP4). The difference between SGP4 and SDP4 is that SDP4 is used for objects with an orbital period above 225 min with SGP4 being for objects with periods below 225 min. The SDP4 algorithm follows the same architecture as the SGP4 algorithm with some added functions and different coefficients, this allows SDP4 to be integrated with a set of sub-functions which are represented on the left side of Figure 2.

## 4. Simulator Performance Metrics

### 4.1. RSO Pixel Position Comparison

To validate the accuracy of the SBOIS to a readily available commercial product such as STK-EOIR, a comparison study using the identical orbital parameters in both simulators was conducted. Images taken by FAI on 15 June 2019, 23:35:50 to 23:36:15 were simulated. This time period was chosen to capture the first sighting of the RADARSAT Constellation Mission (RCM) spacecraft after they were launched on 12 June 2019. Their distinct formation of three spacecraft, combined with other reference stars within FAI’s field of view help not only in determining the pixel locations but also in identifying and correcting for potential lens distortions since it forms a geometrical shape rather than a single data point. Three stars from the Bright Star V5 catalog—#6111, #6158, and #6195—were used as references to prevent errors caused by discrepancies in the simulation of the attitude of the observer spacecraft CASSIOPE. They were chosen for their unique in-line formation that is almost parallel to the RCM formation during the observation period used in this study.

Figure 2 illustrates a set of images from FAI (left), SBOIS (center), and STK-EOIR (right). Highlighted in the red ellipses are the RCM constellation imaged on 15 June. Yellow ellipses depict the locations of three reference stars (Bright Star V5 #6111, #6158, #6195). The RCM, a three-satellite constellation providing daily revisits of Canadian land and ocean, was launched three days prior to the study period; thus, not fully deployed and still retains its distinctive formation. The formation remains visible within FAI’s field of view for 32 s, with a total of 31 images taken.

Table 2 illustrates sample data with three satellite locations with corresponding errors in pixels defined as the difference between FAI images and simulated images from SBOIS and STK-EOIR, respectively. More examples of the pixel locations and errors are listed in Appendix A.

The comparison results are very promising. Pixel accuracy for RCM satellites are mostly within 5 pixels for SBOIS. This error can be attributed to slight position and attitude discrepancies between the real host satellite and the host within the simulator. Observing satellite (CASSIOPE satellite that hosts FAI) ephemeris, for example, plays a critical role in presenting RSO orbit in simulated images. The error of RSO position, as seen from a CCD, is minimal compared to the effect of host satellite position. While the shape and size of the objects affect the brightness of the object, the position of the RSO in the starfield is mostly defined by the accuracy of the RSO position, observer satellite position, and attitude of the observer satellite. RSO attitude and facets (surfaces or shape of the RSO) contribute to the light curve in the form of brightness variation; however, they do not affect the position of the RSO on the simulated images. For the purpose of this study, it was assumed that all space objects are 10-m diameter with a perfect sphere for visibility on the simulated image. The spheres are estimated to have 0.2-reflectance across FAI’s effective spectral band. Statistics, such as light reflectance of materials used, the physical size of the satellite, solar panel placement, and orientation could also affect the centroid of the reflected light perceived by another sensor; but only on a subpixel level. The comparison study between SBOIS and STK-EOIR did not consider the above-listed physical parameters. Instead, all objects were assumed to be of the same size for comparison and to eliminate potential errors which may occur from the continuous attitude changes. STK’s SGP4 propagator was used for the RSO positioning to keep the propagation method the same between SBOIS and STK.

The pixel locations of RCM satellites and reference stars, as shown in Figure 3, were compared by extracting their pixel location from both simulators; then, by calculating differences in the X- and Y-axes. The data from SBOIS are extracted directly from the simulator, as there is a built-in function to the output pixel location. STK, on the other hand, has no such functions to export a file with pixel locations of objects within the field of view of a sensor. It does have a details window which can be viewed via EOIR synthetic scene window, listing the information within each pixel as the user selects it. The information includes X and Y coordinates, objects within the pixel, and object distance from the sensor. The lack of batch-export ability in STK significantly hinders any future analysis if the user does not know the general location of the target at the time period, and likely have to cross-reference with the 3D scenario window in STK to determine the pixels the objects are in before analyzing them.

The reference stars are compared in a similar manner. Since both simulators use the same star catalog (Bright Star V5), significant differences between the two simulators’ outputs were not expected nor observed. It is worth noting that on rare occasions—at 23:36:15, for example—the scale of the star formation seems to be altered slightly (pixel differences between first and last star is different on both simulators), with STK showing a formation that is slightly larger than that of SBOIS. Also, noteworthy that in the last 2 images, as the satellites move towards the edge of the image, the lens distortion effect of the FAI sensor caused the image to warp, decreasing the accuracy of both simulators compared to the real image, as seen in the comparison shown in Appendix A.

As stated in the RCM analysis, SBOIS images depict stars with 6.6 and RSO’s with 5.1-pixel accuracy, comparable to the images generated from STK (mean difference of 6.5 compared to original FAI image. Note that we reported accuracy of 9.4 pixels using the original simulator without the sub-pixel feature [21]. As the simulator was used only to examine the feasibility of RSO identification using low-resolution, the reported pixel accuracy was considered acceptable. With the improvement made since then, we now represent on-orbit images with better than 5.1-pixel accuracy, making the simulator pertinent for various applications. In the current study, we focus on the validity of the simulator in creating a large dataset for AI algorithm training where accuracy and variance of the simulated data is key characteristic. Regardless of the approach, we take in developing AI for this purpose, simulated data needs to represent the real dataset with as much accuracy and flexibility as we can afford.

It is also noted that in comparison with SBOIS, STK-EOIR does not have a built-in sub-pixel output option; if one needs to identify the exact location of an object, it can only be done through a third-party algorithm that calculates the centroid of a multi-pixel shape. This may lead to some albeit small, but frequent inaccuracies in future studies.

### 4.2. Formatting of Mathematical Components

Currently, there are over 20,000 in-orbit objects such as satellites, debris, rockets, revolving around Earth in various orbits, and this number is expected to rise by about an order in magnitude with the newly constructed US space fence coming operational [22]. To account for the large scale of propagation required a Parallel SGP4 method (PSGP4) as outlined in [20] was implemented. In addition, the Bright Star catalog adds around 10,000 objects in the form of stars into the scenario. Combining all the objects in any simulation is a challenging task, as well as the synthetic scene generation required to simulate the field of view and images of a sensor installed on a spacecraft. To successfully complete given tasks, a simulator needs to be capable of simulating the propagation of a great amount of RSOs, while taking the Bright Star Catalog, or any chosen catalog, into account when generating a simulated scene for a sensor; this should be done in a way that does not require significant computation time. This requires the simulator to be optimized, robust, flexible, and customizable. We compare SBOIS and STK-EOIR in the above-listed aspect, with emphasis on computation speed, since SBOIS utilizes such methods in its computation. It is worth noting that STK has the advantage of having a graphical user interface (GUI) and a visual 3D scenario, but these can take valuable computational resources away from the actual simulation; this can be a burden if the user does not require the visualization nor the GUI.

The comparison study was designed with both simulators generating 200 and 1000 images, then comparing their processing time. These results can be seen in Table 3. The results from SBOIS were collected via an output function built into the simulator akin to other auxiliary data, which gives them time to generate the specified time period. The initialization time for SBOIS is between 0.13 to 0.15 s—insignificant for this comparison. Results from STK were taken by subtracting the image creation time of the 1st image from the last image generated by checking the option "auto-generate scene bitmap", which gives the in-between time of the two images. It is worth noting that the size of the EOIR window affects STK calculation speed, but closing it will also stop the generation of images. Therefore, for the following comparisons, all windows were adjusted to the smallest dimension possible to achieve the best results.

The data above compares the computational speed of STK12.0.1 and its compatible EOIR to SBOIS in terms of total time to generate a specified number of images for each comparison category. From the comparison, it is clear that SBOIS is superior to STK in the processing time required under the circumstances that are tested. Apart from the total time needed, the average time taken for each image was also compared. The time needed for SBOIS increases alongside the number of images within a set. the average time per image for a 200-image sequence was 0.34 s and decreased to 0.28 s when the number of images was increased to 1000. STK takes significantly longer than SBOIS to produce the images in both cases with the images produced not being automatically labeled. This implies that SBOIS is more computationally efficient for generating large labeled datasets.

As mentioned earlier, the 3D visualization capability and the GUI of STK help the user greatly in many scenarios but can slow down the program significantly if a large number of objects are active at the same time. To test the limitations of STK, the entire GPS constellation and some independent satellites were imported into a scenario. STK crashed several times during the import process after there are over 60 objects in the scenario. Moreover, the import process can be time-consuming for the user if TLE or other satellite files are not readily available. On average, it takes 1.5 to 2 s to import an object from the AGI database via GUI. Importing all objects within SBOIS into an STK scenario can be extremely time-consuming. Although object import time can be reduced significantly when considering the best-case scenario when the user has satellite orbit files ready, the possibility of STK crashing during the import is still present.

## 5. RSO Detection Algorithms Using Simulated Images

Detection algorithms are out of the scope of this paper, with the goal just to provide simulated data for the training and verification of these algorithms. Here some of the different types of RSO detection algorithms are mentioned and how simulated images are used with them. Firstly, RSO detection algorithms are used to determine if an RSO is in the image, as well as, distinguish between stars and RSO’s. The main way of determining the difference between stars and RSO’s is from the motion that they produce in the image, with stars streaking due to the motion of the camera and host platform. The streaking of the RSO’s is due to the motion of the camera, host platform, and RSO’s motion. When accounting for the sensor and platform motion the only objects that should be moving are the RSO, allowing for determination between the stars and RSO’s. For more information on different detection and centroiding methods, such as machine learning and analytical algorithms see [21,23,24]. For the training of machine learning detection algorithms, hundreds to thousands of images and image sequences are required, which are not always available for space-based platforms. To make up for the lack of data simulated images can be used to train these algorithms. One example of detection algorithms being trained of SBOIS simulated images is presented in Dave’s AMOS technical presentation [5].

## 6. Conclusions

In this paper, we described the architecture of the custom design starfield simulator, SBOIS, and compared its performance with the real images as well as the simulated images generated using a commercial product, STK-EOIR. In comparing FAI images taken on 15 June 2019, 23:35:50 to 23:36:15 where a distinct RADARSAT Constellation Mission (RCM) formation was observed, an average of approximately 5-pixel accuracy was observed in both simulators, demonstrating very similar performance in generating realistic starfield images. SBIOS, however, features an annotation function that each image generated has labeled data for the simulated objects (both RSOs and stars) with position, velocity, pixel centroid, and SNR. In comparing the processing time, SBOIS outperforms for 200 and 1000-image cases with less than 1/3 of the processing time to generate the images.

As part of future algorithm development, we are extracting light curve information of RSOs to estimate the target range and improve the accuracy of the temporal classifier with further characterization parameters. This will be similar to the light curve inversion methods outlined in [6,25,26,27] to estimate an unresolved RSO’s shape, attitude, and optical properties. An additional area of research is the point spread function modeling of sensors using convolution filters from the proposed classification algorithm. The algorithm proposed for PSF modeling would improve the accuracy of the simulator in recreating images and reduce centroiding and classification for future sensor simulations. Performing a similar analysis on a varying range of image datasets from ground and space, from varying sensor types and resolutions could also widen the application of the simulator.

## Figures and Tables

**Figure 1 sensors-21-07868-f001:**
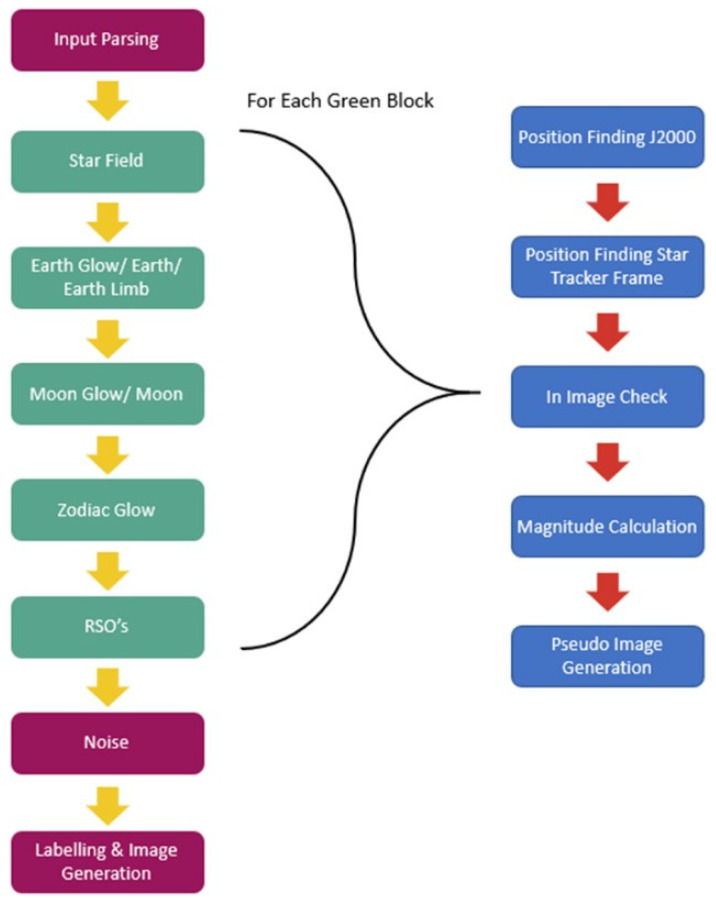
Space-based optical image simulator (SBIOS) system architecture is in green and purple. The image generation sequence for steps 2 through 6 are shown in blue. noisesources include: read-out noise, shot noise, and the point spread function; pseudoimage refers to the simulated image that contains the single light source, namely Earth glow, moon, zodiac glow, and RSO.

**Figure 2 sensors-21-07868-f002:**
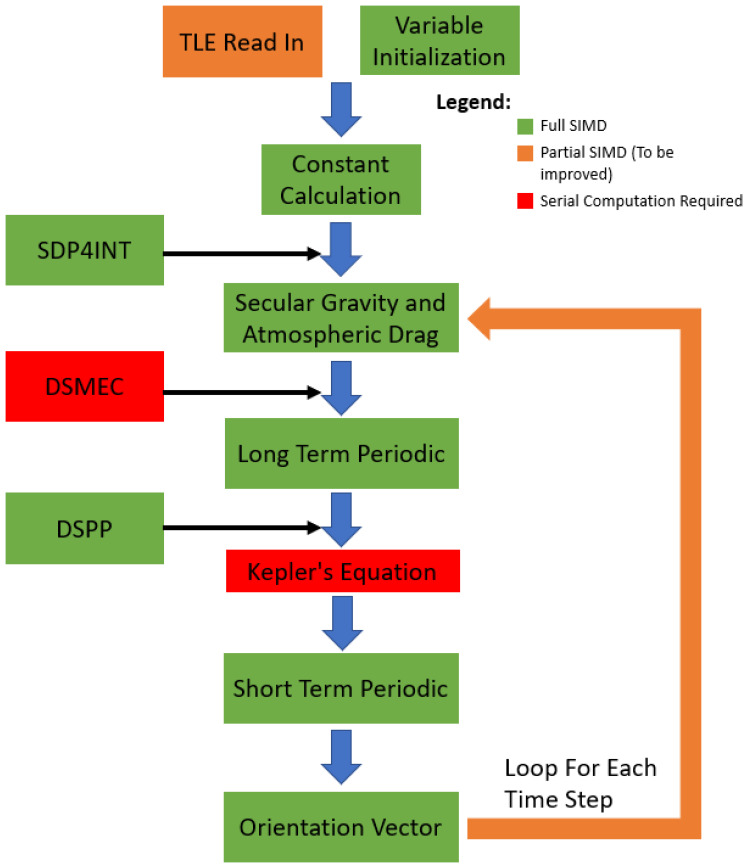
PSGP4 Algorithm Block Diagram, where the color represents the different implementations of SIMD processing. On the left of the main flow, the additional PSDP4 functions are shown with the location of the implementation.

**Figure 3 sensors-21-07868-f003:**
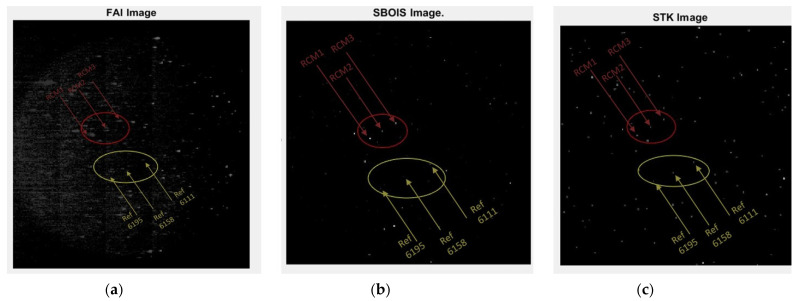
Sample Images from: (**a**) FAI on-orbit observation [15]; (**b**) Simulated using SBOIS (**c**) Simulated using STK-EOIR.

**Table 1 sensors-21-07868-t001:** Fast Aurora Imager Parameters [11].

Parameter	Value
Pixels	256 × 256
Quantum Efficiency, QE	0.66
Optical Transmittance Loss, *τ*	0.9
Aperture Diameter	1.7 cm *
Pixel Size	26 µm
Effective Focal Length	1.38 cm *
Integration Time	0.1 s

* The focal length of the FAI sensor is 68.9 mm and it has an f-number of f/4. This gives it an aperture of 17 mm.

**Table 2 sensors-21-07868-t002:** Summary of Simulator Accuracy.

Objects	SBOIS		STK-EOIR		SBOIS	ESTK
	Row(X)	Col(Y)	Row(X)	Col(Y)	Error	Error
15 June 2019 23:35:50						
Ref 6111	143.0	151.2	143.6	151.0	(1.1, −8.6)	(1.7, −8.8)
Ref 6158	121.6	161.3	120.6	160.7	(−4.4, −5)	(−5.4, −5.6)
Ref 6195	101.1	170.2	100	170.0	(−1.5, −4.8)	(−2.6, −5)
RCM-1	22.6	155.0	13.7	152.6	(3.8, 5.1)	(−5.1, 2.7)
RCM-2	35.5	147.8	30.0	144.6	(1.3, 4.8)	(−4.2, 1.6)
RCM-3	49.9	139.6	45.4	136.6	(−0.3, 4)	(−4.8, 1)

**Table 3 sensors-21-07868-t003:** Computation Time Comparison.

	Processing Time with 200 Image Sequence	Average Time per Image	Processing Time with 1000-Image Sequence	Average Time per Image
SBOIS	67 s	0.34 s	280 s	0.28 s
STK	681 s	3.4 s	3322 s	3.3 s

## Data Availability

The data presented in this study are available on request from the corresponding author. The data is not publicly available due to continuing research and containing sensitive data.

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
