# Peer review of "Simulation of RSO Images for Space Situation Awareness (SSA) Using Parallel Processing"

_sensors, 2021, doi:10.3390/s21237868_

Round 1
Reviewer 1 Report
This paper is a revised version of a previous submission on the SBOIS (space-based optical image simulator) imager. The first submission intended to address a broader panorama while the second is a more focused paper.
The second version has indeed improved from the first proposal. We can now find the details of the parallel processing architecture (figure 2) and both simulation and performance results are concrete allowing a better understanding of the research work.
Still, I have some remarks:
a) The paper mentions (line 3) the existence of AI-based technologies to identify RSOs from ground and space-based observation. Yet, the introduction does not provide any survey/discussion of such methods. The paper must deep into the related work.
b) Format:
Lines 59, 93: send the websites to the reference section.
97: To combined these --> To combine these
105: equations 1 to 2 --> equations 1 and 2
147: respectfully --> respectively
172: was design --> was designed
178: reflected off of the --> reflected off the
187: was also design --> was also designed
191: with there position --> with their position
Conclusions should not be listed in a Table.
Reviewer 2 Report
Only a suggestion and an easy question are added in this round: 1) It may be better to further provide the expression of PSF in Equation (2); 2) Line 127, Line 143-144: Is Dn corresponding to DN?
Author Response
Please see the attachment

This manuscript is a resubmission of an earlier submission. The following is a list of the peer review reports and author responses from that submission.
Round 1
Reviewer 1 Report
The FAI image in figure 2, could be better, there is much noise, is not easy to detect the RCM satellites.
In the 256 row there is a "down" repeated.
Please add some more examples of FAI and simulated images with SBOIS and STK and compare the results with the commercial simulator.
Reviewer 2 Report
This paper introduces the SBOIS (space-based optical image simulator), a spaceborne imager that generates resident space objects (RSO) from low-resolution images. The SBOIS architecture is based on FAI (fast auroral imager) parameters. The performance metrics addressed in this paper are the RSO centroids’ coordinates, the implementation of parallel processing for optimal computation, and the robustness in simulation parameters. To verify its performance, such metrics were compared to those obtained with the STK-EOIR simulator. Results show a similar performance with the advantage of an automatic data labelling data for SBOIS.
Overall, the paper is confusing. As it can be understood, SBOIS targets to detect RSO, locate them in the image, and generate 3D models of their shapes all this using parallel processing:
1) The paper does not address RSO detection/identification at all. How to discriminate stars/satellites/debris/any other RSO is not detailed nor exemplified. The paper points to reference [1]. Detailing performance metrics without presenting the simulator’s main purpose and performance has little significance.
2) The paper promises to present the implementation of the parallel processing structure. The paper just points to the SGP4 method (line 218) with no further mention to the topic.
3) Similarly, I missed to find the “robustness of the simulation parameters”. Parameters in Table 1 are related to the FAI sensor. Why are they robust?
4) Figure 1 is not explained. In addition to the text, an image example illustrating the process sequence will be helpful.
Minor:
5) Line 16: FAI needs to be defined here
Reviewer 3 Report
The title of the paper is ‘Space Situational Awareness (SSA) Data Simulation Using Parallel Processing’, but in the text, I really cannot find any description on using parallel processing. In the Simulator Architecture section, I do not find how to implement the space-based optical image simulator (SBOIS) using Parallel Processing. Consider “2. Simulator Architecture”, “3. Simulator Performance Metrics”, and “4. Application of Space-Based Optical Image Simulator (SBOIS) for SSA Studies”, I think that the structure of the paper is confused, and the topic of the paper is unclear, so that the content is inconsistent to the title of the paper. According to the abstract, the authors try to present an overview of star field simulator to generate realistic representation of images from spaceborne imagers, and particularly focus on low-resolution images such as those taken with commercial grade star tracker that contain various RSO in one images. I don’t think an overview is enough for a research paper. According to the abstract, the authors said that the accuracy and computational efficiency are compared to commercial simulator, namely STK-EOIR to demonstrate the performance of the simulator. Whose accuracy and computational efficiency are compared to commercial simulator, namely STK-EOIR? Some other problems, such as: 1) Line 16: what is FAI? what is CASSIOPE mission? 2) It seems that section 3.2 was lost; 3) Line 218: what is Parallel SGP4?, etc.